

# Entanglement, non-hermiticity and duality

**Li-Mei Chen[1], Shuai A. Chen[2⋆] and Peng Ye[1†]**

**1** School of Physics and State Key Laboratory of Optoelectronic Materials and Technologies,
Sun Yat-sen University, Guangzhou, 510275, China
**2** Institute for Advanced Study, Tsinghua University, Beijing, 100084, China

⋆ s-chen16@mails.tsinghua.edu.cn,   † yepeng5@mail.sysu.edu.cn

## Abstract

Usually duality process keeps energy spectrum invariant. In this paper, we provide a duality, which keeps entanglement spectrum invariant, in order to diagnose quantum entanglement of non-Hermitian non-interacting fermionic systems. We limit our attention to non-Hermitian systems with a complete set of biorthonormal eigenvectors and an entirely real energy spectrum. The original system has a reduced density matrix $\rho_o$ and the real space is partitioned via a projecting operator $\mathcal{R}_o$. After dualization, we obtain a new reduced density matrix $\rho_d$ and a new real space projector $\mathcal{R}_d$. Remarkably, entanglement spectrum and entanglement entropy keep invariant. Inspired by the duality, we defined two types of non-Hermitian models, upon $\mathcal{R}_o$ is given. In type-I exemplified by the "non-reciprocal model", there exists at least one duality such that $\rho_d$ is Hermitian. In other words, entanglement information of type-I non-Hermitian models with a given $\mathcal{R}_o$ is entirely controlled by Hermitian models with $\mathcal{R}_d$. As a result, we are allowed to apply known results of Hermitian systems to efficiently obtain entanglement properties of type-I models. On the other hand, the duals of type-II models, exemplified by "non-Hermitian Su-Schrieffer-Heeger model", are always non-Hermitian. For the practical purpose, the duality provides a potentially *efficient* computation route to entanglement of non-Hermitian systems. Via connecting different models, the duality also sheds lights on either trivial or nontrivial role of non-Hermiticity played in quantum entanglement, paving the way to potentially systematic classification and characterization of non-Hermitian systems from the entanglement perspective.

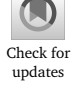

# 1  Introduction

For the past decades, Hermitian quantum matters have been intensively investigated. Classification and characterization of Hermitian quantum matters are deeply rooted in many-body treatment on quantum entanglement. Without symmetry, gapped phases are classified into short-ranged entangled (SRE) and long-range entangled (LRE) phases [1, 2]. LRE phases, such as fractional quantum Hall states [3] are physically characterized by the robust ground state degeneracy on closed manifold and braiding statistics of topological excitations. Such phases are often called *intrinsic* topological order [4–8]. LRE phases cannot be adiabatically connected to a direct product state via a local unitary transformation (LU) that attempts to disentangle local degrees of freedom. In contrast to LRE, there exists *at least* one LU transformation such that SRE states can be connected to the direct product state without crossing phase transitions. When symmetry is considered, both LRE and SRE have finer phase structures. Symmetry Protected Topological phases, e.g., the Haldane phase [9, 10], are symmetric SRE states that admit symmetry-protected boundary anomaly [6, 11–14]. On the other hand, symmetric LRE states are called Symmetry Enriched Topological phases [6, 15–17] that admit fractionalized quantum number carried by topological excitations. Inspired by quantum information, by partitioning the real-space $X$ into two subregions: $X_o = \mathcal{A}_o \cup \mathcal{B}_o$, quantum entanglement between the two subregions can be quantitatively measured via von Neumann entanglement entropy (EE): $S_{EE} = -\mathrm{Tr}\rho_o \log \rho_o$ with $\rho_o =: e^{-h_o^E}$ being a reduced density matrix of the subregion $\mathcal{A}_o$ [18]. The full spectrum of the entanglement "Hamiltonian" $h^E$, known as entanglement spectrum (ES) [19] encodes more fruitful information about quantum entanglement. In short, it has been well recognized that EE and ES can help identify and distinguish universal properties of phases [2, 4, 20–22].

On the other hand, Hermiticity of a Hamiltonian is one of the key postulates of isolated quantum systems in order to ensure both probability conservation and the real-valuedness of eigen-energies. Nevertheless, non-Hermiticity is still physically relevant and ubiquitous in, e.g., open systems. Non-Hermitian physics provides a versatile platform for a variety of classical and quantum systems with concrete lattice models such as non-reciprocal model, non-Hermitian SSH model [23–32].

For non-Hermitian systems, while there has been tremendous progress in many aspects, the entanglement information is far less known, compared to the progress on many-body entanglement of Hermitian quantum matters such as aforementioned LRE and SRE. One may ask whether or not the introduction of non-Hermiticity can substantially reshape universal behaviors of entanglement properties of Hermitian systems [18, 33–35]. One may alterna-

tively ask whether or not there exist non-Hermitian systems whose ES can be computed from well-studied Hermitian systems? In other words, non-Hermiticity in such systems is irrelevant in the quantum entanglement. Finally, is it possible to unify non-Hermitian and Hermitian systems from the entanglement perspective?

While general correlated systems are difficult, let us focus on non-Hermitian non-interacting systems with Hamiltonian $H_o$[1] [36–42]. Then we can construct a reduced density matrix $\rho_o$ by pairs of right and left eigenstates of $H_o$ [43]. More specifically, the entanglement Hamiltonian of $H_o$[2], denoted by $h_o^E$, is analytically determined by $h_o^E = \log[(\mathcal{R}_o \mathcal{P}_o \mathcal{R}_o)^{-1} - \mathbb{I}]$, where two operators $\mathcal{R}_o$ and $\mathcal{P}_o$ ($\mathcal{R}_o^2 = \mathcal{R}_o, \mathcal{R}_o^\dagger = \mathcal{R}_o, \mathcal{P}_o^2 = \mathcal{P}_o, \mathcal{P}_o^\dagger \neq \mathcal{P}_o$) impose quantum-state projections onto the subregion $\mathcal{A}_o$ of $X_o$ and occupied eigenstates of $H_o$, respectively. It should be noted that the Fock-space projector $\mathcal{P}_o$ is no longer Hermitian, but the real-space projector $\mathcal{R}_o$, by definition, must always be Hermitian.

In this paper, we build a rigorous duality between a non-Hermitian non-interacting Hamiltonian $H_o$ and its dual Hamiltonian, denoted by $H_d$. Remarkably, $H_o$ and $H_d$ share the same ES, i.e., $\text{Spec}(h_o^E) = \text{Spec}(h_d^E)$.[3] Meanwhile, the dual Hamiltonian $H_d$ and its reduced density matrix $\rho_d$ may be either non-Hermitian or Hermitian. By means of this duality, we establish exotic connections between different models, regardless of Hermiticity. Before moving to detailed technical discussions to appear in the main text, let us concisely illustrate the duality here via Fig. 1. In Fig. 1, the duality consists of two key steps. Step-①, a similarity transformation

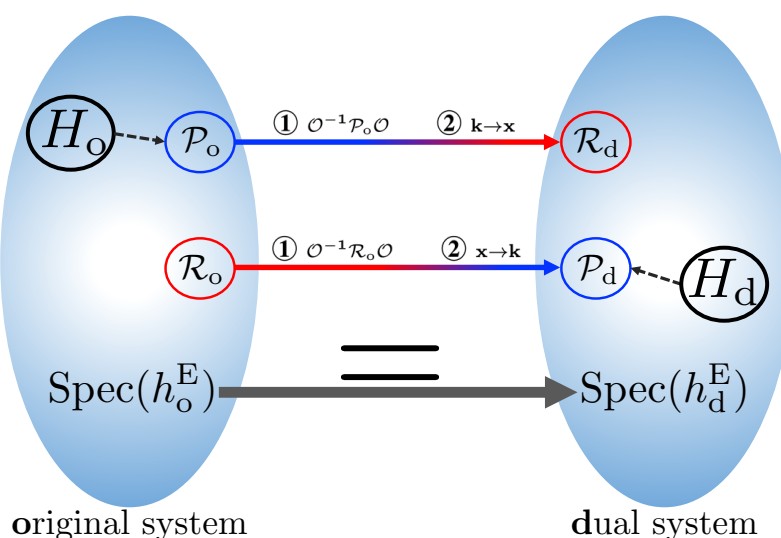

Figure 1: Schematic illustration of the duality. The duality between the original system $H_o$ and the dual system $H_d$ is split into two steps. In the first step ①, we introduce a similarity transformation on both Fock-space projector $\mathcal{P}_o$ and real-space projector $\mathcal{R}_o$. In the second step ②, we interchange real space and momentum space. Entanglement spectrum and entanglement entropy, which can be computed from diagonalizing reduced density matrices as in Eq. (10), keep invariant. Details can be found in the main text.

$\mathcal{O}$ is applied to *not only* $\mathcal{P}_o$ *but also* $\mathcal{R}_o$. The feature of '*simultaneously acting on both projectors*' is very crucial and will be elaborated in the main text. Step-②, the real space and Fock-space are exchanged. As a result, two new projectors $\mathcal{R}_d$ and $\mathcal{P}_d$ of the dual system are naturally

---

[1]If such non-Hermitian systems are realized as meanfield Hamiltonians of correlated systems, the results in this paper are also applicable.

[2]Unless otherwise stated, Hamiltonians of non-Hermitian systems in this paper are assumed to act on the Hilbert space with a complete set of biorthonormal eigenvectors and possess an entirely real spectrum.

[3]Here the symbol $\text{Spec}(\mathcal{O})$ denotes the spectrum of the operator $\mathcal{O}$.

defined. Both projectors enter $h_{\mathrm{d}}^{\mathrm{E}}$ in the standard way: $h_{\mathrm{d}}^{\mathrm{E}} = \log[(\mathcal{R}_{\mathrm{d}}\mathcal{P}_{\mathrm{d}}\mathcal{R}_{\mathrm{d}})^{-1} - \mathbb{I}]$. Since $\mathcal{R}_{\mathrm{d}}$ is interpreted as a real-space projection in the dual system, its Hermiticity must be guaranteed: $\mathcal{R}_{\mathrm{d}} = \mathcal{R}_{\mathrm{d}}^{\dagger}$. Therefore, we require that the equality $\Theta\mathcal{P}_{\mathrm{o}}^{\dagger} = \mathcal{P}_{\mathrm{o}}\Theta$ always holds, where $\Theta := \mathcal{O}\mathcal{O}^{\dagger}$. Theorem 1 of the main text will be introduced to guarantee $\mathrm{Spec}(\mathcal{R}_{\mathrm{o}}\mathcal{P}_{\mathrm{o}}\mathcal{R}_{\mathrm{o}}) = \mathrm{Spec}(\mathcal{R}_{\mathrm{d}}\mathcal{P}_{\mathrm{d}}\mathcal{R}_{\mathrm{d}})$. Therefore, the duality in Fig. 1 keeps ES unaffected, i.e., $\mathrm{Spec}(h_{\mathrm{o}}^{\mathrm{E}}) = \mathrm{Spec}(h_{\mathrm{d}}^{\mathrm{E}})$, which completes the duality process.

The duality has profound consequences. Physically, the duality inspires us to divide non-Hermitian Hamiltonians into two types, namely, type-I and type-II, when a real-space partition $\mathcal{R}_{\mathrm{o}}$ is given. In each case of type-I, there exists at least one duality process such that $\mathcal{P}_{\mathrm{d}}^{\dagger} = \mathcal{P}_{\mathrm{d}}$ and thus $\rho_{\mathrm{d}}^{\dagger} = \rho_{\mathrm{d}}$. Therefore, in type-I, ES can be fully determined by its Hermitian dual, i.e., $H_{\mathrm{d}}$. On the contrary, in type-II, it is impossible to compute ES from known results of Hermitian systems through the present duality, since all $H_{\mathrm{d}}$'s are non-Hermitian. Mathematically, whether $\rho_{\mathrm{d}}$ is Hermitian or not essentially relies on the commutator $[\Theta, \mathcal{R}_{\mathrm{o}}]$. If there exists an $\mathcal{O}$ such that $[\Theta, \mathcal{R}_{\mathrm{o}}] = 0$, then $\rho_{\mathrm{d}}$ is Hermitian and $H_{\mathrm{o}}$ is of type-I. On the one hand, from the definition, non-Hermiticity of type-I doesn't play any essential role in EE and ES. But on the other hand, for the practical purpose, we are allowed to *efficiently* compute ES and EE of type-I systems by means of known results of Hermitian systems. To demonstrate it, the non-reciprocal lattice model is identified as type-I. As a byproduct, we prove that ES and EE of this model are independent of the parameter $\alpha$ [Eq. (26)] that measures the degree of "non-reciprocality". In contrast to type-I, the dual system $H_{\mathrm{d}}$ is always non-Hermitian for type-II systems. It indicates that entanglement information of type-II system cannot be understood through any known results of Hermitian systems in the present duality process. The non-Hermitian Su-Schrieffer-Heeger (SSH) model is one of simplest examples of type-II. In Table 1, we list the criteria and entanglement properties of the two types of systems.

The remainder of this paper is organized as follows. In Sec. 2, some useful facts on non-Hermitian quantum physics are reviewed. We explain the duality process by presenting one theorem and two corollaries in Sec. 3. Two typical examples (nonreciprocal model and non-Hermitian SSH model) are computed in details in Sec. 4. In Sec. 5, this work is concluded with several remarks and future directions. Appendices include further supplemental information on the duality.

## 2 Preliminaries

For a non-Hermitian system of free fermions, its second-quantized Hamiltonian can be written as

$$H_{\mathrm{o}} = \sum_{\alpha\beta} c_{\alpha}^{\dagger} H_{\alpha\beta} c_{\beta}, \tag{1}$$

where $H_{\mathrm{o}} \neq H_{\mathrm{o}}^{\dagger}$ and fermionic operators $c_{\alpha}^{\dagger}$ and $c_{\alpha}$ satisfy the standard anticommunication relations $\{c_{\alpha}^{\dagger}, c_{\beta}\} = \delta_{\alpha\beta}$. Suppose $H_{\mathrm{o}}$ admits a complete set of biorthonormal eigenvectors $\{|r,\alpha\rangle, |l,\alpha\rangle\}$ that satisfy

$$\langle l,\alpha|r,\beta\rangle = \delta_{\alpha\beta}, \quad \sum_{\alpha} |l,\alpha\rangle\langle r,\alpha| = \mathbb{I}, \tag{2}$$

with $|r,\alpha\rangle$ and $|l,\alpha\rangle$ being the right and left eigenvectors,

$$H_{\mathrm{o}}|r,\alpha\rangle = \epsilon_{\alpha}|r,\alpha\rangle, \quad H_{\mathrm{o}}^{\dagger}|l,\alpha\rangle = \epsilon_{\alpha}^{*}|l,\alpha\rangle. \tag{3}$$

Here $\alpha$ is the spectral label, $\delta_{\alpha\beta}$ denotes the Kronecker delta function and $\mathbb{I}$ is the identity matrix. Therefore, we have a spectral decomposition

$$H_{\mathrm{o}} = \sum_{\alpha} \epsilon_{\alpha}|r,\alpha\rangle\langle l,\alpha|. \tag{4}$$

By introducing bifermionic operators

$$\psi_{r\alpha}^\dagger |0\rangle \equiv |r,\alpha\rangle , \quad \psi_{l\alpha}^\dagger |0\rangle \equiv |l,\alpha\rangle , \tag{5}$$

with the anti-commutation relations $\{\psi_{l\alpha}, \psi_{r\beta}^\dagger\} = \delta_{\alpha\beta}$, we can straightforwardly construct a many-body state,

$$|G_r\rangle = \prod_{\alpha \in occ} \psi_{r\alpha}^\dagger |0\rangle , \quad |G_l\rangle = \prod_{\alpha \in occ} \psi_{l\alpha}^\dagger |0\rangle , \tag{6}$$

where occ denotes a set of the occupied states. *Hereafter, in the rest of the paper, unless otherwise stated, a non-Hermitian Hamiltonian is assumed to act on the Hilbert space with a complete set of biorthonormal eigenvectors and possess an entire real energy spectrum.* Mathematically, it is equivalent to the condition that [38, 39] there is an invertible operator $\mathcal{O}$ such that

$$H_o \Theta = \Theta H_o^\dagger , \tag{7}$$

with

$$\Theta := \mathcal{O}\mathcal{O}^\dagger . \tag{8}$$

From the right and left states, a density matrix can be constructed [43] as

$$\rho = |G_r\rangle\langle G_l| , \tag{9}$$

such that $\rho^2 = \rho$ and $\rho^\dagger \neq \rho$. With this generalized notation, provided a partition on the real-space into subregions $X_o = \mathcal{A}_o \cup \mathcal{B}_o$, we realize measurement of entanglement $S_{EE} = -\mathrm{Tr}\rho_o \ln \rho_o$, where the reduced density matrix $\rho_o$ is defined by taking partial trace of subsystem $\mathcal{B}_o$,

$$\rho_o = \mathrm{Tr}_{\mathcal{B}_o}\rho =: e^{-h_o^E}, \quad h_o^E = \sum_{i,j \in \mathcal{A}_o} c_i^\dagger (h_E)_{ij} c_j . \tag{10}$$

The entanglement Hamiltonian $h_o^E$ is introduced [19] in Eq. (10) whose spectrum $\mathrm{Spec}(h_o^E)$ encodes more fruitful information on quantum entanglement. For a non-interacting system, entanglement Hamiltonian $h_o^E$ is uniquely determined by a two-point correlation matrix $C_o$ with elements $(C_o)_{ij} = \langle G_l| c_i^\dagger c_j |G_r\rangle$, $i, j \in \mathcal{A}_o$ via a relation

$$h_o^E = \log\left(C_o^{-1} - \mathbb{I}\right) , \tag{11}$$

with $\mathbb{I}$ being an identity matrix [44, 45]. Furthermore, we can reformulate $C_o$ as

$$C_o = \mathcal{R}_o \mathcal{P}_o \mathcal{R}_o , \tag{12}$$

in terms of the real-space projector

$$\mathcal{R}_o = \sum_{i \in \mathcal{A}_o} |i\rangle\langle i| , \tag{13}$$

onto region $\mathcal{A}_o$ and Fock-space projector

$$\mathcal{P}_o = \sum_{\alpha \in occ} |r,\alpha\rangle\langle l,\alpha| , \tag{14}$$

onto occupied states [43, 46]. One significant feature is that $\mathcal{P}_o$ is no longer Hermitian, namely, $\mathcal{P}_o \neq \mathcal{P}_o^\dagger$, while the real-space projector $\mathcal{R}_o$ is, by definition, Hermitian $\mathcal{R}_o = \mathcal{R}_o^\dagger$. In our duality [see Fig. 1], the basic notations for the dual system can be obtained by replacing the subscript index o with d. For example, the reduced density matrix $\rho_d$ for the dual system defines its entanglement Hamiltonian $h_d^E$ via $\rho_d =: e^{-h_d^E}$ and other formula work in the same way.

# 3 Duality

When entanglement meets non-Hermiticity, how are the universal behaviours of entanglement reshaped? Alternatively, is it possible that we can comprehend entanglement of non-Hermitian systems based on the knowledge of Hermitian systems? For this purpose, we propose a duality, which is depicted in Fig. 1. This duality process keeps ES and EE unaffected and leads to two different types of non-Hermitian models.

## 3.1 Duality process

As schematically illustrated in Fig. 1, our duality is conducted by two steps. In the first step, under a similarity transformation, $\mathcal{P}_o$ is mapped to a Hermitian operator, and, *simultaneously* $\mathcal{R}_o$ is mapped to an operator that may or may not be Hermitian. We exchange the roles of momenta and positions in the second step and obtain projectors $\mathcal{P}_d$ and $\mathcal{R}_d$ in a dual system. Since a projector $\mathcal{R}_d$ must be Hermitian $\mathcal{R}_d^\dagger = \mathcal{R}_d$ in order to depict a real-space partition, we have to impose an requirement on the similarity transformation $\mathcal{O}$ in the first step,

$$\Theta \mathcal{P}_o^\dagger = \mathcal{P}_o \Theta, \tag{15}$$

with $\Theta = \mathcal{O}\mathcal{O}^\dagger$ defined in Eq. (8). Besides, invariance of ES and EE requires a condition

$$\mathrm{Spec}(\mathcal{R}_o \mathcal{P}_o \mathcal{R}_o) = \mathrm{Spec}(\mathcal{R}_d \mathcal{P}_d \mathcal{R}_d), \tag{16}$$

as indicated by Eq. (11). The two conditions in Eqs. (15) and (16) can be satisfied as indicated by Theorem 1.

**Theorem 1** *Given a Hamiltonian $H_o$ acting on the Hilbert space with a complete set of biorthonormal eigenvectors with an entirely real spectrum, there exists an invertible similarity transformation $\mathfrak{A} = \mathcal{O}^{-1}\mathcal{P}_o\mathcal{O}$ and $\mathfrak{B} = \mathcal{O}^{-1}\mathcal{R}_o\mathcal{O}$ such that*

$$\mathrm{Spec}(\mathcal{R}_o \mathcal{P}_o \mathcal{R}_o) = \mathrm{Spec}(\mathfrak{A}\mathfrak{B}\mathfrak{A}), \tag{17}$$

*with $\mathfrak{A} = \mathfrak{A}^\dagger$. Here the symbol $\mathrm{Spec}(\mathcal{O})$ in Eq. (17) denotes spectrum of $\mathcal{O}$.*

Theorem 1 states that there is always an invertible similarity transformation such that a rearrangement on projectors keep the spectrum unaffected, which is a generalized version of the Hermitian counterpart [46, 47]. We can exchange roles of momenta and positions, in the second step. We re-interpret $\mathfrak{B}$ as a new Fock-space projector, *re-denoted as $\mathcal{P}_d$*, to describe occupied states and $\mathfrak{A}$, as a new real-space projector, *re-denoted as $\mathcal{R}_d$* to depict the real partition. Thus, we obtain a correlation matrix $C_d = \mathcal{R}_d \mathcal{P}_d \mathcal{R}_d$ from which we can design the dual Hamiltonian $H_d$ with an entirely real spectrum. Theorem 1 along with the formula in Eq. (11) then indicates that $H_d$ shares the same EE and ES with $H_o$. At this stage, we finish building our duality between two systems, $H_o$ with a real-space partition $\mathcal{R}_o$ and $H_d$ with a real-space partition $\mathcal{R}_d$. The procedures are depicted in Fig. 1. Such a duality allows us to diagnose entanglement properties of a non-Hermitian system $H_o$ from the computation in its dual one.

Below we give a proof of Theorem 1.

**Proof 3.1** *From the property of $H_o$, there is an invertible operator $\mathcal{O}$ with $\Theta := \mathcal{O}\mathcal{O}^\dagger$ that satisfies the condition in Eq. (7). Then, we have $\Theta \mathcal{P}_o^\dagger = \mathcal{P}_o \Theta$, or equivalently,*

$$\mathcal{O}^{-1}\mathcal{P}_o\mathcal{O} = \mathcal{O}^\dagger \mathcal{P}_o^\dagger \mathcal{O}^{\dagger -1}. \tag{18}$$

*We define* $\mathfrak{A} = \mathcal{O}^{-1}\mathcal{P}_o\mathcal{O}$ *and* $\mathfrak{A}$ *is Hermitian* $\mathfrak{A} = \mathfrak{A}^\dagger$. *Since an invertible similarity transformation does not alter spectrum, we have*

$$\text{Spec}(\mathcal{R}_o\mathcal{P}_o\mathcal{R}_o) = \text{Spec}(\mathcal{O}\mathfrak{B}\mathfrak{A}\mathfrak{B}\mathcal{O}^{-1}) = \text{Spec}(\mathfrak{B}\mathfrak{A}\mathfrak{B}), \tag{19}$$

*where* $\mathfrak{B} = \mathcal{O}^{-1}\mathcal{R}_o\mathcal{O}$ *may be either Hermitian or not.*

*The next is to prove* $\text{Spec}(\mathfrak{B}\mathfrak{A}\mathfrak{B}) = \text{Spec}(\mathfrak{A}\mathfrak{B}\mathfrak{A})$. *Suppose an eigenstate* $|\xi\rangle$ *of* $\mathfrak{B}\mathfrak{A}\mathfrak{B}$ *with*

$$\mathfrak{B}\mathfrak{A}\mathfrak{B}|\xi\rangle = c|\xi\rangle, \tag{20}$$

*and then* $\mathfrak{B}|\xi\rangle = |\xi\rangle$ *by observing*

$$c\mathfrak{B}|\xi\rangle = \mathfrak{B}\mathfrak{B}\mathfrak{A}\mathfrak{B}|\xi\rangle = c|\xi\rangle. \tag{21}$$

*Thus,* $\mathfrak{A}|\xi\rangle$ *is an eigenstate of* $\mathfrak{A}\mathfrak{B}\mathfrak{A}$:

$$\mathfrak{A}\mathfrak{B}\mathfrak{A}(\mathfrak{A}|\xi\rangle) = c(\mathfrak{A}|\xi\rangle). \tag{22}$$

*Therefore, given an eigenstate* $|\xi\rangle$ *of* $\mathfrak{B}\mathfrak{A}\mathfrak{B}$ *with eigenvalue c,* $\mathfrak{A}|\xi\rangle$ *is an eigenstate of* $\mathfrak{A}\mathfrak{B}\mathfrak{A}$ *with eigenvalue c. The converse is also true. Finally, we have*

$$\text{Spec}(\mathcal{R}_o\mathcal{P}_o\mathcal{R}_o) = \text{Spec}(\mathfrak{B}\mathfrak{A}\mathfrak{B}) = \text{Spec}(\mathfrak{A}\mathfrak{B}\mathfrak{A}). \tag{23}$$

## 3.2 Two types of non-Hermitian systems

The duality shown in Fig. 1 maps a non-Hermitian system $H_o$ into a new one $H_d$ while they share the same ES and EE. Therefore, we can diagnose entanglement in $H_o$ by means of $H_d$. If the dual system $H_d$ turns out to be Hermitian, we can assert that non-Hermiticity indeed does not play any essential role in entanglement of such a system $H_o$. The condition for $H_d$ being Hermitian, i.e.,

$$\mathcal{O}^{-1}\mathcal{R}_o\mathcal{O} = \mathcal{O}^\dagger\mathcal{R}_o\mathcal{O}^{\dagger-1}, \tag{24}$$

is that a similarity transformation $\mathcal{O}$ exists such that $\Theta$ commutes with $\mathcal{R}_o$,

$$[\Theta, \mathcal{R}_o] = 0. \tag{25}$$

Consequently, given $H_o$ if at least such a similarity transformation exists to meet Eq. (25), we regard such a system as type-I. Otherwise, the system is categorized into type-II. Obviously, a Hermitian Hamiltonian belongs to type-I, for which we can simply take $\mathcal{O}$ to be an identity matrix. The ES and EE of type-I obey the same tendency as a Hermitian system. Thus we can understand it within the context of Hermitian systems. For example, we expect that the entanglement formula inspired by Wisdom conjecture [48–51] still work and EE and ES are directly obtained from known results on Hermitian systems without complicated calculations. On the other hand, non-Hermiticity is supposed to play an intrinsic role in entanglement of type-II.

The operator $\Theta$ varies for different choices on $\mathcal{O}$. Explicitly, given $\mathcal{O}_1$ that satisfies Eq. (7), the operator $\mathcal{O}_2 = \mathcal{O}_1 U_1 S U_2$ also meets Eq. (7), but $\Theta$ is changed. Here $U_1$ denotes a unitary transformation that diagonalizes $\mathcal{O}_1 H_o \mathcal{O}_1^{-1}$, $U_2$ is an arbitrary unitary matrix and $S$ is an invertible matrix that commutes with spectral matrix of $H_o$. In Appdendix A, we give an example to illustrate this point. In practice, to determine the type of a system, one can start with $\mathcal{O}$ that diagonalizes $\mathcal{O}^{-1}H_o\mathcal{O} = \Lambda$. If $[\mathcal{O}\mathcal{O}^\dagger, \mathcal{R}_o] = 0$, then it belongs to type-I. Otherwise, we have to check whether some $S$ exists to solve the equation $[\mathcal{O}SS^\dagger\mathcal{O}^\dagger, \mathcal{R}_o] = 0$. In Appendix A, we make more explanation on the procedure to determine the type of a given non-Hermitian model.

Based on the theorem, two corollaries naturally follow.

Table 1: Criteria and entanglement spectrum (ES) for categorizations on non-Hermitian free systems as well as two examples. We obtain two types of non-Hermitian free systems according to whether at least one $\Theta$ in Eq. (8) exists to commute with a real space partition $\mathcal{R}_o$.

|  | Criteria | ES | Example |
|---|---|---|---|
| Type-I | At least one $\Theta$ commutes with $\mathcal{R}_o$ | Real | Non-reciprocal model [Sec. 4.1] |
| Type-II | No $\Theta$ commutes with $\mathcal{R}_o$ | Real or complex | non-Hermitian SSH model [Sec. 4.2] |

**Corollary 1** *The ES for type-I non-Hermitian system is real. A non-Hermitian system with complex ES belongs to type-II.*

The Corollary 1 is a direct consequence of the duality process. The real-valued ES of a type-I arises from the identical spectrum to a Hermitian system. The converse-negative proposition of the first part produces the second argument. We point out that we do not exclude a type-II system possessing a real ES.

**Corollary 2** *In the case of a Hermitian system, one recovers the "position-momentum duality".*

The Corollary 2 is obvious since one can simply choose $\mathcal{O}$ to be an identity matrix, which exactly recovers the result established in Ref. [46]. In Table 1, we summarize the properties of two types of non-Hermitian systems as well as two typical examples that will be presented in Sec. 4.1 and Sec. (4.2).

# 4 Examples

In this section, we present two examples to exemplify the two types of non-Hermitian systems.

## 4.1 An example for Type-I: Nonreciprocal model

As a warm-up, we consider one of the simplest non-Hermitian models on a chain of $L$ sites

$$H_o = -t \sum_{x=1}^{L} \left( e^\alpha c_x^\dagger c_{x+1} + e^{-\alpha} c_{x+1}^\dagger c_x \right), \qquad (26)$$

where $c_x^\dagger$ and $c_x$ are the fermion creation and annihilation operators at site $x$, respectively. The nonreciprocal left/right hopping $te^{\pm\alpha}$ can arise from asymmetric gain/loss, which is shown in Fig. 2. Under an open boundary condition (OBC), it is exactly solvable and one can write down the right and left eigenvectors $|r,k\rangle$ and $|l,k\rangle$ as

$$|r,k\rangle = \sqrt{\frac{2}{L+1}} \sum_{x=1}^{L} e^{-\alpha x} \sin \frac{\pi k x}{L+1} |x\rangle, \qquad (27)$$

$$|l,k\rangle = \sqrt{\frac{2}{L+1}} \sum_{x=1}^{L} e^{\alpha x} \sin \frac{\pi k x}{L+1} |x\rangle, \qquad (28)$$

with a real gapless spectrum $\epsilon_o(k) = -2t \cos \frac{\pi k}{L+1}$ parametrized by momentum indices $k = 1, \cdots, L$.

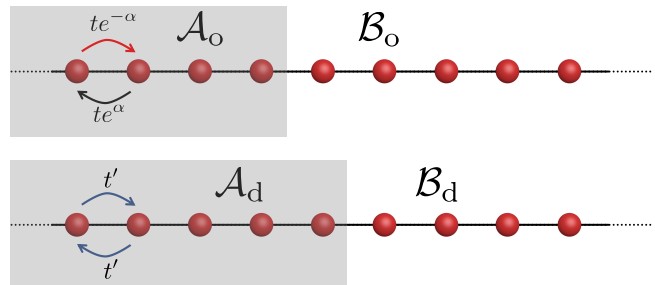

Figure 2: Illustration of non-reciprocal model and its dual system for a general filling and partition. After duality, the partition is changed from $X_o = \mathcal{A}_o \cup \mathcal{B}_o$ to $X_d = \mathcal{A}_d \cup \mathcal{B}_d$. Non-Hermiticity in $H_o$ arises from non-reciprocal hopping $te^{\pm\alpha}$ while the dual system is Hermitian with hopping integral $t'$.

Now we conduct the duality in Fig. 1. In **Step**-①, we can choose $\mathcal{O}$ to be

$$\mathcal{O} = \mathrm{diag}(e^{-\alpha}, \cdots, e^{-L\alpha}), \tag{29}$$

which describes the transformation

$$c_x^\dagger \to c_x^\dagger e^{-x\alpha}, \quad c_x \to c_x e^{x\alpha}. \tag{30}$$

Partition the system into two subregions $\mathcal{A}_o$ and $\mathcal{B}_o$ with the Fock-space and real-space projectors being

$$\mathcal{P}_o = \sum_{k\in\mathrm{occ}} |r,k\rangle\langle l,k|, \mathcal{R}_o = \sum_{x\in\mathcal{A}_o} |x\rangle\langle x|. \tag{31}$$

After Step-① with a similarity transformation in Eq. (29) acting on both $\mathcal{P}_o$ and $\mathcal{R}_o$ in Eq. (31),

$$\mathcal{O}^{-1}\mathcal{P}_o\mathcal{O} = \sum_{k\in\mathrm{occ}} |k\rangle\langle k|, \mathcal{O}^{-1}\mathcal{R}_o\mathcal{O} = \sum_{x\in\mathcal{A}_0} |x\rangle\langle x|, \tag{32}$$

in **Step**-②, we interchange the roles of momenta and positions, which defines dual projectors

$$\mathcal{R}_d = \sum_{x\in\mathrm{occ}} |x\rangle\langle x|, \mathcal{P}_d = \sum_{k\in\mathcal{A}_o} |k\rangle\langle k|, \tag{33}$$

where the two basis's $|k\rangle$ and $|x\rangle$ in $\mathcal{R}_d$ and $\mathcal{P}_d$ satisfy the relation, $|k\rangle = \sqrt{\frac{2}{L+1}}\sum_{x=1}^{L} \sin\frac{\pi k x}{L+1}|x\rangle$. Here the Fock-space projector $\mathcal{P}_d$ means that states with momentum in region $\mathcal{A}_o$ are occupied and $\mathcal{R}_d$ defines a real-space partition. For example, at half-filling of $H_o$, we can conduct a partition with $\mathcal{A}_o$ containing half the chain. Then we can take the spectral dispersion to be $\epsilon_d = -2t'\cos\frac{\pi k x}{L+1}$, and the dual Hamiltonian $H_d$ reads

$$H_d = -t'\sum_{x=1}^{L} \left(c_x^\dagger c_{x+1} + c_{x+1}^\dagger c_x\right), \tag{34}$$

with a partition $\mathcal{A}_d$ being half the chain [4]. For a general partition, we can introduce a chemical potential such that $\epsilon_d(k) < 0$ when $k \in \mathcal{A}_o$. In Fig. 2, we depict lattices for $H_o$ and its dual $H_d$ with partition $X_o = \mathcal{A}_o \cup \mathcal{B}_o$ and $X_d = \mathcal{A}_d \cup \mathcal{B}_d$ respectively.

[4]Here the strength of hopping integral $t'$ does not influence entanglement.

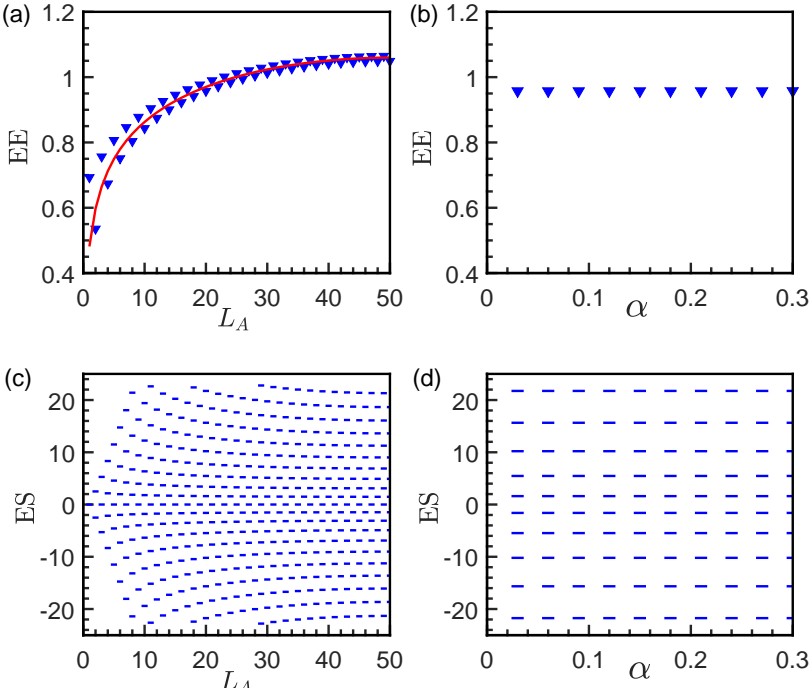

Figure 3: Entanglement entropy $S_{EE}$ (a) *v.s.* $L_{\mathcal{A}_o}$ with $L = 100$, $\alpha = 0.1$ and (b) *v.s.* $\alpha$ with $L = 100$, $L_{\mathcal{A}_o} = 20$; Entanglement spectrum (c) *v.s.* $L_{\mathcal{A}_o}$ with $L = 100$, $\alpha = 0.1$ and (d) *v.s.* $\alpha$ with $L = 100$, $L_{\mathcal{A}_o} = 20$ for non-reciprocal model in Eq. (26). In (a), $S_{EE}$ calculated via the definition in Eq. (10) marked by blue triangles is consistent with the formula of $S_{EE}$ guided by a red line that is inferred by the duality. (b) shows $S_{EE}$ is independent of $\alpha$. (c) depicts EE dependence on the size of region $\mathcal{A}_o$. (d) shows EE is not altered when $\alpha$ changes. All calculations are conducted under open boundary condition at half-filling with $t = 1$.

Remarkably, the dual Hamiltonian in Eq. (34) does not depend on the parameter $\alpha$, which indicates that non-Hermiticity plays no role in EE and ES in original non-Hermitian system. It is further numerically checked in Fig. 3(b) and Fig. 3(d) that no changes in EE and ES occur when we change $\alpha$. Furthermore, the quantity $\mathcal{O}\mathcal{O}^\dagger$ for the present choice is diagonal in the real-space such that at least one dual system is Hermitian regardless of the system partition.

In summary, the non-Hermitian non-reciprocal model in Eq. (26) shares the same ES and EE with the dual Hermitian free fermi gas in Eq. (34), which exemplifies a type-I system. This feature allows us to understand entanglement of type-I under the framework of a Hermitian system. For example, instead of complicated calculation, we can directly extract that in the nonreciprocal model, momenta and positions enter $S_{EE}$ symmetrically with the expression [46, 52–55] as

$$S_{EE} = \frac{1}{6} \ln \left[ \frac{L}{\pi} \sin \frac{\pi L_{\mathcal{A}}}{L} \sin \frac{\pi L_F}{L} \right] + \cdots , \tag{35}$$

where $L_{\mathcal{A}}$ denotes the length of region $\mathcal{A}_o$, $L_F$ is the number of occupied bands of its $L$ eigenstates and $\cdots$ includes constant and finite-size corrections from $1/L$ and higher. Figure 3(a) shows consistence between $S_{EE}$ from Eq. (10) and the duality.

## 4.2 An example for Type-II: Non-Hermitian SSH model

Another example is the non-Hermitian Su-Schrieffer-Heeger (SSH) model in a bipartite lattice with $2N$ sites at half-filling,

$$
\begin{aligned}
H_{\mathrm{o}} = \sum_{x=1}^{N-1} & \left( \omega c_{2x}^{\dagger} c_{2x+1} + v c_{2x+1}^{\dagger} c_{2x+2} \right) + \text{h.c.} \\
& + \sum_{x=1}^{N} i u \left( c_{2x}^{\dagger} c_{2x} - c_{2x+1}^{\dagger} c_{2x+1} \right),
\end{aligned}
\tag{36}
$$

with $u, v, \omega \in \mathbb{R}$. Here we introduce staggered imaginary chemical potentials. Under a periodic boundary condition (PBC), then $H_{\mathrm{SSH}}$ is translationally invariant and can be reformulated in the Fock-space (i.e., momentum space) as

$$
H_{\mathrm{o}} = \oplus_k H_k \quad \text{with} \quad H_k = \begin{bmatrix} iu & v_k \\ v_k^* & -iu \end{bmatrix},
\tag{37}
$$

where $v_k = \omega e^{-ika} + v$ with $a$ the lattice constant. The system is $\mathcal{PT}$-invariant $\sigma_x H_k \sigma_x = H_k^*$ with spectrum $\epsilon_{k,\pm} = \pm \sqrt{|v + w e^{ika}|^2 - u^2}$ and we restrict ourselves in the region of real spectrum.

To conduct the duality, in **Step-①**, we choose the matrix $\mathcal{O} = \oplus_k \mathcal{O}_k$ in the momentum space to be

$$
\mathcal{O}_k = \begin{bmatrix} -e^{-i(\theta_k + \varphi_k)} & 1 \\ 1 & e^{-i(\theta_k - \varphi_k)} \end{bmatrix},
\tag{38}
$$

where we reparametrize

$$
\rho_k e^{i\varphi_k} = v + \omega e^{ika}, \quad \rho_k e^{i\theta_k} = \epsilon_{k,+} + iu.
\tag{39}
$$

At half-filling, we have the two projectors

$$
\mathcal{P}_{\mathrm{o}} = \sum_{k \in \text{occ}} |r, k, -\rangle \langle l, k, -|, \quad \mathcal{R}_{\mathrm{o}} = \sum_{i \in \mathcal{A}_{\mathrm{o}}} \sum_{s=1,2} |i, s\rangle \langle i, s|,
\tag{40}
$$

where $\mathcal{P}_{\mathrm{o}}$ projects onto all occupied states with $|r, k, -\rangle$ ($|l, k, -\rangle$) as its right(left) eigenvector and $\mathcal{R}_{\mathrm{o}}$ defines partition with $s = 1, 2$ labeling two sublattices. After a similarity transformation acting on both the two projectors in Eq. (40), in **Step-②**, we interchange the roles on positions and momenta and we have

$$
\mathcal{P}_{\mathrm{d}} = \sum_{k \in \mathcal{A}_{\mathrm{o}}, \ell=\pm} \mathcal{O}_{\mathrm{d}}^{-1} |k, \ell\rangle \langle k, \ell| \mathcal{O}_{\mathrm{d}},
\tag{41}
$$

$$
\mathcal{R}_{\mathrm{d}} = \sum_{x \in \text{occ}} |x, -\rangle \langle x, -|,
\tag{42}
$$

where in the dual real-space, $\mathcal{O}_{\mathrm{d}} = \oplus_{x=1}^{N} \mathcal{O}_x$ is the similarity transformation in the first step and $\ell$ can be interpreted as internal degrees or layer indices. The expression of $\mathcal{O}_x$ is obtained by replacing $k$ in Eq. (38) by $x$, and

$$
\rho_x e^{i\varphi_x} = v + \omega e^{i\frac{2\pi x}{Na}}, \rho_x e^{i\theta_x} = \epsilon_x + iu.
\tag{43}
$$

We can introduce the following biorthogonal eigenvectors

$$
|r, k, \ell\rangle = \mathcal{O}_{\mathrm{d}}^{-1} |k, \ell\rangle,
\tag{44}
$$

and bifermionic operators

$$\psi^\dagger_{r,k,\ell}|0\rangle = |r,k,\ell\rangle , \quad \psi^\dagger_{l,k,\ell}|0\rangle = |l,k,\ell\rangle , \tag{45}$$

such that

$$\mathcal{P}_{\rm d} = \sum_{k\in\mathcal{A}_{\rm o},\ell=\pm} |r,k,\ell\rangle\langle l,k,\ell| , \tag{46}$$

and the dual Hamiltonian takes the form as $H_{\rm d} = \sum_{k,s} \epsilon_{k,\ell}\psi^\dagger_{r,k,\ell}\psi_{l,k,\ell}$ where $\epsilon_{k,\ell}$ is the dispersion relation that is constrained by $\epsilon_{k,\ell} < 0$ for $k\in\mathcal{A}_{\rm o}$, $\ell = \pm$. Specifically, we can take $\mathcal{A}_{\rm o}$ to be half the chain, then the dispersion relation can be simply chosen as $\epsilon_{k,\ell} = -2t\sqrt{N}\cos ka$. Thus, the dual Hamiltonian can be formulated as

$$H_{\rm d} = -t\sum_x \sum_{y=x\pm a} c^\dagger_x e^{i\mathbf{A}_{x,y}\cdot\sigma+iA^0_{x,y}\sigma_0}c_y , \tag{47}$$

where $c_x = (c_{x,-},c_{x,+})^{\rm T}$ is a two-component spinor, $\sigma = (\sigma_x,\sigma_y,\sigma_z)$ is a vector of Pauli matrices and $\sigma_0$ is an identity matrix. The fields $\mathbf{A}_{x,y}$ and $A^0_{x,y}$ reside at the link $(x,y)$ and no longer keeps anti-symmetric on its spatial indices, $\mathbf{A}_{x,y} \neq -\mathbf{A}_{y,x}, A^0_{x,y} \neq -A^0_{y,x}$. Thus we map non-Hermitian SSH model to non-Hermitian non-Abelian gauge field theory. The original band indices are interpreted as component indices and $\mathcal{R}_{\rm d}$ involves partition on the internal spinor degrees as well as spatial degrees. General expressions for the dual Hamiltonian are presented in Appendix B. On the other hand, commutation with $\mathcal{R}_{\rm o}$ requires $\Theta$ to be a block diagonal matrix and no $S$ exists to make $\mathcal{O}SS^\dagger\mathcal{O}^\dagger$ commute with $\mathcal{R}_{\rm o}$. Therefore, the non-Hermitian SSH belongs to type-II, which is consistent with its complex ES.

At the end of this section, we give some useful remarks on our duality and Dyson map. Our duality is distinct from a Dyson map [38,56]. A Dyson map is referred to as a similarity transformation which maps a non-Hermitian system to a Hermitian counterpart, which keeps an energy spectrum unchanged but generically generally alters entanglement spectrum. In contrast, our duality has a different mission: keeping entanglement spectrum unchanged while imposing no constraints on energy spectrum. For this purpose, we have designed the duality shown in Fig. 1 which includes two necessary steps. For a $\mathcal{PT}$-symmetric system, one can define a parity operator to act on the Hamiltonian just as the way as $\Theta$ in Eq. (8). However, the factorization constraint $\Theta = \mathcal{O}\mathcal{O}^\dagger$ does not allow $\Theta$ to be such a parity operator. In fact, our duality is applicable to a general non-Hermitian system since as shown in Fig. 1 the duality targets at invariance of ES and EE and no restriction is needed for energy spectra.

## 5 Conclusions

In this paper, we are interested in the role that non-Hermiticity plays in quantum entanglement and develop a rigorous duality for probing the role. We make an initial step towards a unified picture of non-Hermiticity and Hermiticity from the entanglement perspective. Explicitly, we have considered non-Hermitian non-interacting systems whose Hamiltonians are assumed to act on the Hilbert space with a complete set of biorthonormal eigenvectors and possess an entirely real spectrum. We classify these systems into two types, which is summarized in Table 1. For type-I, non-Hermiticity plays no role and thus we can efficiently obtain entanglement entropy and entanglement spectrum by means of well-studied results in Hermitian systems. For type-II, non-Hermiticity indeed plays an intrinsic role. Several implications and applications are discussed.

Motivated by this work, we present here several questions for future study. First, is there a similar/generalized duality or generalized LU for characterizing many-body entanglement

of non-Hermitian interacting systems [57, 58]? For example, it is important to define non-Hermitian version of LRE states via generalized LU. Second, is it possible to find Wannier interpolation on non-Hermitian non-interacting entanglement [59]? Such an interpolation may help us to semi-analytically understand ES of non-Hermitian fermion systems. Third, how can we further physically distinguish two subclasses of type-II systems? In Corollary 1, we have shown that ES of type-II systems may be either complex or real. So, it is interesting to further investigate finer structures of type-II systems. Third, a free Hamiltonian plays a role as a mean-field theory of an interacting system. What is the relation between two interacting systems if their mean-field theory are dual to each other? Fourth, it is appealing to generalize our duality into momentum space entanglement [60, 61]. Furthermore, it is important to perform experimental measurement to distinguish entanglement behaviors of the two-type non-Hermitian systems using the experimental breakthroughs [62, 63] and in particular to confirm our statement for type-I non-Hermitian systems. After the first arxiv version, we were aware that there are some other interesting investigations on entanglement of non-Hermitian systems, such as Refs. [64–66]. In the future, it will be interesting to combine these increasing new findings and duality together.

## Acknowledgments

This work was supported in part by the Sun Yat-sen University startup grant, Guangdong Basic and Applied Basic Research Foundation under Grant No. 2020B1515120100, National Natural Science Foundation of China (NSFC) Grant (No. 11847608 & No. 12074438).

In the appendix, we make some explanation on the choices of $\mathcal{O}$ in the first step of the duality and detailed derivations on the dual model for non-Hermitian SSH model.

## A   A two-site model

In this appendix, We concentrate on the procedures to determine the type of a given non-Hermitian system. As is shown in Fig. 1, our duality is split into two steps. In Step-①, a similarity transformation $\mathcal{O}$ are conducted on both Fock-space and real-space projectors $\mathcal{P}_o$ and $\mathcal{R}_o$ and then in Step-② an interchange between interpretation on momenta and positions follows. According the duality, we classify non-Hermitian systems into two types. Explicitly, if at least one Hermitian dual system $H_d$ exists, a non-Hermitian system belongs to type-I. Otherwise, it belongs to type-II. In practice, we need to check all possible $\mathcal{O}$ in Step-①.

Here we consider a system with only two lattice sites that reads

$$H_o = re^{i\theta}c_1^\dagger c_1 + sc_1^\dagger c_2 + re^{-i\theta}c_2^\dagger c_2 + sc_2^\dagger c_1, \tag{48}$$

where $c_s$ ($s = 1, 2$) is a fermion annihilation operator at the site $s$. The system has only one particle. Non-Hermiticity arises from a complex-valued chemical potential. Following the preliminary, we introduce bifermionic operators

$$
\begin{aligned}
\psi_{l-} &= \frac{1}{\sqrt{2}\cos\alpha}(e^{i\alpha}c_1 + c_2), \\
\psi_{l+} &= -\frac{1}{\sqrt{2}\cos\alpha}(-c_1 + e^{i\alpha}c_2), \\
\psi_{r-}^\dagger &= \frac{1}{\sqrt{2}}(c_1^\dagger + e^{-i\alpha}c_2^\dagger), \\
\psi_{r+}^\dagger &= \frac{1}{\sqrt{2}}(-e^{-i\alpha}c_1^\dagger + c_2^\dagger),
\end{aligned}
\tag{49}
$$

with $se^{i\alpha} = ir\sin\theta + \sqrt{s^2 - r^2\sin^2\theta}$. And then we have the spectral decomposition

$$H_{\rm o} = \epsilon_-\psi^\dagger_{r-}\psi_{l-} + \epsilon_+\psi^\dagger_{r+}\psi_{l+} \, , \tag{50}$$

where $\epsilon_\pm = r\cos\theta \pm \sqrt{s^2 - r^2\sin^2\theta}$ are eigenenergies of the two states $|r,\pm\rangle = \psi^\dagger_{r\pm}|0\rangle$. The ground state describes occupation of the state $|r,-\rangle$ and the corresponding Fock-space projector $\mathcal{P}_{\rm o}$ is

$$\mathcal{P}_{\rm o} = \psi^\dagger_{r-}|0\rangle\langle 0|\psi_{l-} \, . \tag{51}$$

We make a partition where the subregion $\mathcal{A}_{\rm o}$ only contains the first site with the real-space projector being

$$\mathcal{R}_{\rm o} = c^\dagger_1|0\rangle\langle 0|c_1 \, . \tag{52}$$

Let's start our duality. In **Step-①**, we can choose $\mathcal{O}$ to be

$$\mathcal{O} = \frac{1}{\sqrt{2}}e^{i\alpha}c^\dagger_1|0\rangle\langle 0|c_1 + \frac{1}{\sqrt{2}}c^\dagger_1|0\rangle\langle 0|c_2 - \frac{1}{\sqrt{2}}c^\dagger_2|0\rangle\langle 0|c_1 + \frac{1}{\sqrt{2}}e^{i\alpha}c^\dagger_2|0\rangle\langle 0|c_2 \, , \tag{53}$$

which defines a similarity transformation on both $\mathcal{P}_{\rm o}$ and $\mathcal{R}_{\rm o}$. We introduce the notations

$$\begin{aligned}
c_\pm &= \mathcal{O}^{-1}\psi_{r\pm}\mathcal{O} \, , \\
c^\dagger_\pm &= \mathcal{O}^{-1}\psi^\dagger_{l\pm}\mathcal{O} \, , \\
\psi_{r1,2} &= \mathcal{O}^{-1}c_{1,2}\mathcal{O} \, , \\
\psi^\dagger_{l1,2} &= \mathcal{O}^{-1}c^\dagger_{1,2}\mathcal{O} \, ,
\end{aligned} \tag{54}$$

under which we get a compacted form after action of $\mathcal{O}$

$$\mathcal{O}^{-1}\mathcal{P}_{\rm o}\mathcal{O} = c^\dagger_-|0\rangle\langle 0|c_-, \quad \mathcal{O}^{-1}\mathcal{R}_{\rm o}\mathcal{O} = \psi^\dagger_{r1}|0\rangle\langle 0|\psi_{l1} \, . \tag{55}$$

It is easy to check that in Eq. (54), $c_\pm$ and $c^\dagger_\pm$ are conventional fermion operators while $\psi_{r1,2}$ and $\psi^\dagger_{l1,2}$ are bifermionic operators [See Sec. 2]. In **Step-②**, we exchange indices $\{\pm\}$ and spatial indices $\{1,2\}$, thus obtaining two projectors $\mathcal{P}_{\rm d}$ and $\mathcal{R}_{\rm d}$ in the dual system

$$\mathcal{P}_{\rm d} = \psi^\dagger_{r-}|0\rangle\langle 0|\psi_{l-}, \quad \mathcal{R}_{\rm d} = c^\dagger_1|0\rangle\langle 0|c_1 \, , \tag{56}$$

where

$$\begin{aligned}
\psi_{l-} &= \frac{1}{\sqrt{2}\cos\alpha}(c_1 - e^{-i\alpha}c_2) \, , \\
\psi_{l+} &= \frac{1}{\sqrt{2}\cos\alpha}(e^{-i\alpha}c_1 + c_2) \, , \\
\psi^\dagger_{r-} &= \frac{1}{\sqrt{2}}(e^{i\alpha}c^\dagger_1 - c^\dagger_2) \, , \\
\psi^\dagger_{r+} &= \frac{1}{\sqrt{2}}(c^\dagger_1 + e^{i\alpha}c^\dagger_2) \, .
\end{aligned} \tag{57}$$

One should not confuse operators in Eq. (56) with those in the original system $H_{\rm o}$. We can build the dual Hamiltonian $H_{\rm d}$ by designing its spectrum $\pm\cos\alpha$,

$$\begin{aligned}
H_{\rm d} &= -\cos\alpha\,\psi^\dagger_{r-}\psi_{l-} + \cos\alpha\,\psi^\dagger_{r+}\psi_{l+} \\
&= -i\sin\alpha\,c^\dagger_1c_1 + c^\dagger_1c_2 + c^\dagger_2c_1 + i\sin\alpha\,c^\dagger_2c_2 \, .
\end{aligned} \tag{58}$$

When $\alpha \neq 0$, non-Hermiticity of $H_d$ arises from the complex-valued chemical potential. Consistently, the quantity $\Theta = \mathcal{O}\mathcal{O}^\dagger$ in Eq. (8)

$$\Theta = 2c_1^\dagger|0\rangle\langle 0|c_1 + 2i\sin\alpha c_1^\dagger|0\rangle\langle 0|c_2 + 2i\sin\alpha c_2^\dagger|0\rangle\langle 0|c_1 + 2c_2^\dagger|0\rangle\langle 0|c_2 , \qquad (59)$$

indeed fails to commute with $\mathcal{R}_o$

$$[\Theta, \mathcal{R}_o] = -2i\sin\alpha c_1^\dagger|0\rangle\langle 0|c_2 + 2i\sin\alpha c_2^\dagger|0\rangle\langle 0|c_1 . \qquad (60)$$

To determine the type of $H_o$, we have to check whether at least one Hermitian dual $H_d$ exists. Suppose

$$S = \lambda_1 c_1^\dagger|0\rangle\langle 0|c_1 + \lambda_2 c_2^\dagger|0\rangle\langle 0|c_2 . \qquad (61)$$

Then $\mathcal{O}S$ also satisfies the requirement in **Step-①**. However, we can not find any $S$ to make $[\mathcal{O}SS^\dagger\mathcal{O}^\dagger, \mathcal{R}_o] = 0$,

$$[\mathcal{O}SS^\dagger\mathcal{O}^\dagger, \mathcal{R}_o] = (e^{-i\alpha}|\lambda_1|^2 - e^{i\alpha}|\lambda_2|^2)c_1^\dagger|0\rangle\langle 0|c_2 - (e^{-i\alpha}|\lambda_1|^2 - e^{i\alpha}|\lambda_2|^2)c_2^\dagger|0\rangle\langle 0|c_1 . \qquad (62)$$

Therefore, $H_o$ belongs to type-II. On the other hand, we can directly calculate the reduced density matrix

$$\rho_o = \text{Tr}_{\mathcal{B}_o}|r,-\rangle\langle l,-| = \frac{1}{2\cos\alpha}(e^{-i\alpha}c_1^\dagger|0\rangle\langle 0|c_1 + e^{i\alpha}|0\rangle\langle 0|) , \qquad (63)$$

and entanglement spectrum is complex-valued and

$$S_{\text{EE}} = -\frac{e^{-i\alpha}}{2\cos\alpha}\log\frac{e^{-i\alpha}}{2\cos\alpha} - \frac{e^{i\alpha}}{2\cos\alpha}\log\frac{e^{i\alpha}}{2\cos\alpha} . \qquad (64)$$

As comparison, we consider non-reciprocal model on a lattices with two sites,

$$H_o = rc_1^\dagger c_1 + t_{12}c_1^\dagger c_2 + rc_2^\dagger c_2 + t_{21}c_2^\dagger c_1 , \qquad (65)$$

with single particle and subregion containing the first site. In step-①, we can choose $\mathcal{O}$ to be

$$\mathcal{O} = \sqrt{t_{12}}c_1^\dagger|0\rangle\langle 0|c_1 + \sqrt{t_{12}}c_1^\dagger|0\rangle\langle 0|c_2 - \sqrt{t_{21}}c_2^\dagger|0\rangle\langle 0|c_1 + \sqrt{t_{21}}c_2^\dagger|0\rangle\langle 0|c_2 . \qquad (66)$$

To determine the type of Hamiltonian in Eq. (65), suppose $S = \lambda_1 c_1^\dagger|0\rangle\langle 0|c_1 + \lambda_2 c_2^\dagger|0\rangle\langle 0|c_2$ and then

$$[\mathcal{O}SS^\dagger\mathcal{O}^\dagger, \mathcal{R}_o] = (|\lambda_1|^2 - |\lambda_2|^2)\sqrt{t_{12}t_{21}}c_1^\dagger|0\rangle\langle 0|c_2 - (|\lambda_1|^2 - |\lambda_2|^2)\sqrt{t_{12}t_{21}}c_2^\dagger|0\rangle\langle 0|c_1 . \qquad (67)$$

When $|\lambda_1| = |\lambda_2|$, $[\mathcal{O}SS^\dagger\mathcal{O}^\dagger, \mathcal{R}_o] = 0$ and we are allowed to build a map to a Hermitian system. When $|\lambda_1| \neq |\lambda_2|$, the dual system is non-Hermitian. Therefore, $H_o$ in Eq. (65) belongs to type-I. One can also follow the steps in Fig. 1 to construct the dual system, which is just like what we do for the system in Eq. (50).

# B   Non-Hermitian SSH model

Here we present details on duality of non-Hermitian SSH model discussed in Sec. 4. To be more transparent, we work in the framework of Dirac's notations. The similarity transformation $\mathcal{O} = \oplus_k \mathcal{O}_k$ can be written as

$$\mathcal{O}_k = -e^{-i(\theta_k+\varphi_k)}|k,1\rangle\langle k,1| + |k,1\rangle\langle k,2| + |k,2\rangle\langle k,1| + e^{-i(\theta_k-\varphi_k)}|k,2\rangle\langle k,2| , \qquad (68)$$

where $|k,s\rangle$ denotes a basis vector carrying momentum $k$ on sublattice $s$. So we have $\mathfrak{B}$,

$$
\begin{aligned}
\mathfrak{B} &= \sum_{x \in \mathcal{A}} \mathcal{O}^{-1}|x,s\rangle\langle x,s|\mathcal{O} \\
&= \frac{1}{N}\sum_{x \in \mathcal{A}}\sum_{k,k'}\mathcal{O}_k^{-1}|x,s\rangle\langle x,s|\mathcal{O}_{k'} \\
&= \frac{1}{N}\sum_{x \in \mathcal{A}}\sum_{k,k'}\frac{e^{i(k-k')x}}{1+e^{-2i\theta_k}}\Big[\big(e^{i\varphi_k-i\varphi_{k'}}e^{-i\theta_k-i\theta_{k'}}+1\big)|k,1\rangle\langle k',1| \\
&\quad + \big(-e^{i\varphi_k}e^{-i\theta_k}+e^{i\varphi_{k'}}e^{-i\theta_{k'}}\big)|k,1\rangle\langle k',2| + \big(e^{-i\varphi_k}e^{-i\theta_k}-e^{-i\varphi_{k'}}e^{-i\theta_{k'}}\big)|k,2\rangle\langle k',1| \\
&\quad + \big(e^{-i\varphi_k+i\varphi_{k'}}e^{-i\theta_k-i\theta_{k'}}+1\big)|k,2\rangle\langle k',2|\Big],
\end{aligned}
\tag{69}
$$

where we use the relation

$$
\mathcal{O}_k|x,1\rangle = \frac{1}{\sqrt{N}}e^{ikx}\left(-e^{-i(\theta_k+\varphi_k)}|k,1\rangle + |k,2\rangle\right)
\tag{70}
$$

$$
\mathcal{O}_k|x,2\rangle = \frac{1}{\sqrt{N}}e^{ikx}\left(|k,1\rangle + e^{-i(\theta_k-\varphi_k)}|k,2\rangle\right),
\tag{71}
$$

with $\rho_k e^{i\varphi_k} = \upsilon + \omega e^{ika}$ and $\rho_k e^{i\theta_k} = \epsilon_k + iu$. Exchange the roles of positions and momenta and we have the dual Fock-space projector $\mathcal{P}_d$ to occupied states

$$
\begin{aligned}
\mathcal{P}_d &= \frac{1}{N}\sum_{x,y,k \in \mathcal{A}}\frac{e^{i(x-y)k}}{1+e^{-2i\theta_x}}\Big[\big(e^{i\varphi_x-i\varphi_y}e^{-i\theta_x-i\theta_y}+1\big)|x,-\rangle\langle y,-| \\
&\quad + \big[-e^{i\varphi_x}e^{-i\theta_x}+e^{i\varphi_y}e^{-i\theta_y}\big]|x,-\rangle\langle y,+| + \big[e^{-i\varphi_x}e^{-i\theta_x}-e^{-i\varphi_y}e^{-i\theta_y}\big]|x,+\rangle\langle y,-| \\
&\quad + \big[e^{-i\varphi_x+\varphi_y}e^{-i\theta_x-i\theta_y}+1\big]|x,+\rangle\langle y,+|.
\end{aligned}
\tag{72}
$$

Therefore, we have the dual Hamiltonian,

$$
H_d = \frac{1}{N}\sum_{x,y,k,\ell,\ell'}\epsilon_{k,\ell}e^{i(x-y)k}U_{xy}^{\ell\ell'}c_{x,\ell}^\dagger c_{y,\ell'},
\tag{73}
$$

where

$$
U_{xy} = \frac{1}{1+e^{-2i\theta_x}}\begin{bmatrix} e^{i\varphi_x-i\varphi_y}e^{-i\theta_x-i\theta_y}+1 & -e^{i\varphi_x}e^{-i\theta_x}+e^{i\varphi_y}e^{-i\theta_y} \\ e^{-i\varphi_x}e^{-i\theta_x}-e^{-i\varphi_y}e^{-i\theta_y} & e^{-i\varphi_x+i\varphi_y}e^{-i\theta_x-i\theta_y}+1 \end{bmatrix},
\tag{74}
$$

and $\rho_x e^{i\varphi_x} = \upsilon + \omega e^{i\frac{2\pi x}{Na}}, \rho_x e^{i\theta_x} = \epsilon_x + iu$. The spectrum dispersion relation is constrained to $\epsilon_{k,\ell} < 0$ when $k \in \mathcal{A}$. Its form depends on choices of region $\mathcal{A}_o$. For example, if $\mathcal{A}_o$ is half the chain, we can take $\epsilon_{k,\ell} = -2t\sqrt{N}\cos ka$ to be independent of index $\ell$. In this case, we arrange $c_x = (c_{x,-}, c_{x,+})^T$ as a two-component spinor such that the dual Hamiltonian can be formulated compactly,

$$
H_d = -t\sum_x\sum_{y=x\pm a}c_x^\dagger U_{xy}c_y = -t\sum_x\sum_{y=x\pm a}c_x^\dagger e^{i\mathbf{A}_{x,y}\cdot\sigma+iA_{x,y}^0\sigma_0}c_y.
\tag{75}
$$

Here we identify $U_{xy} = e^{i\mathbf{A}_{x,y}\cdot\sigma+iA_{x,y}^0\sigma_0}$ to give non-Abelian gauge field theory of the non-Hermitian version. The form of $\epsilon_{k,\ell}$ for a general region $\mathcal{A}_o$ can be obtained by adding a proper chemical potential.

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
