# Peer review of "Entanglement, Non-Hermiticity, and Duality"

_SciPost Physics, doi:SciPost Phys. 11, 003 (2021)_

## Round 1 · Referee Report · Anonymous (Referee 1) · 2021-2-25

Strengths

1) The paper tries to systematize the treatment of non-unitary quantum mechanical systems in relation to computations of entanglement. This is a worthy effort, and I am not aware of other approaches were this is done so systematically. As such it opens a new pathway for research in this area. 2) The paper is quite well written and many examples are presented. 3) The paper elaborates further on the connection between non-unitarity and entanglement, establishing links between two areas of research that tend to be studied separately and from quite different viewpoints.

Weaknesses

1) The paper does not cite a lot of relevant literature, both in the context of non-hermitian quantum mechanics and that of entanglement in non-unitary systems. 2) The paper is not sufficiently clear about the novelty of some of the results, particularly the duality that the authors seem to highlight as one of their main contributions looks very similar to some standard results in the context of non-hermitian quantum mechanics. 3) Connected to the above, the authors should make a bigger effort to emphasize what is standard knowledge and what is new in their approach. 4) Finally, I have some questions regarding the interpretation of some of their results. I will elaborate further in my report.

Report

In my view, the authors make an interesting contribution to two areas of study that have traditionally developed quite independently: on the one hand the study of non-hermitian quantum mechanical systems and their interpretation and on the other hand, the study of entanglement measures and also their interpretation.

The paper proposes a systematic procedure to relate a non-hermitian quantum system to either a hermitian counterpart or another non-hermitian model. They terms this connection a duality and show (including some theorems) that the entanglement entropies of a system and its dual are the same.

The paper includes many examples and through those examples is quite clear about the systematic procedure that one must follow in order to compute the entanglement entropy of the dual system associated with a non-hermitian system.

The paper is generally well written and very explicit about the computations that are performed.

Overall, I think the paper is of sufficient quality and novelty as to deserve to be published in SciPost but it requires a bit of revision and clarification of the main result. I summarize my main suggestions/questions in the "Requested changes" section.

Requested changes

1) My first observation is that the paper only cites a very small number of works on non-hermitian quantum mechanics (references 33,34). While these works are certainly relevant, they fit within a much larger and long-standing field which deserves to be acknowledge more. In particular, the references below should be included too:

The idea that quantum mechanical systems with non-Hermitian hamiltonians could still describe sensible physics was first considered by Wigner in the classical paper:

Wigner E 1960 Normal form of antiunitary operators J. Math. Phys. 1 409–13.

But much of the current interest in non-Hermitian quantum mechanical systems was sparked by the seminal paper

Bender C M and Boettcher S 1998 Real spectra in Non-Hermitian Hamiltonians having PT symmetry Phys. Rev. Lett. 80 5243–6

Where a simple non-hermitian Hamiltonian was shown to have real eigenvalues for a certain parameter range and the results was related to the presence of Parity and Time-Reversal symmetries (PT).

Since then, the field of PT-symmetric quantum mechanics has become huge and there is a lot of activity. Some good reviews are:

Bender C M 2007 Making sense of non-Hermitian Hamiltonians Rep. Prog. Phys. 70 947–1018 Mostafazadeh A 2008 Pseudo Hermitian quantum mechanics, Int. J. Geom. Meth. Mod. Phys. 7, 1191-1306 (2010).

As well as the recent book:

PT-symmetry in quantum and classical physics, Bender, Carl M, Dorey, Patrick E, Dunning, Clare, Fring, Andreas, Hook, Daniel W, Jones, Hugh F, Kuzhel, Sergii, Lévai, Géza & Tateo, Roberto (2018).

Then, more widely, in the context of CFT and critical quantum spin chains, there have been also some results regarding the Rényi and EE of non-unitary models. A proposal for such a computation was given in:

Bianchini, Castro-Alvaredo, Doyon, Levi and Ravanini, Entanglement Entropy of Non Unitary Conformal Field Theory, J. Phys A48 (2015) 4 A4FT01.

Using corner transfer matrix methods, a similar result was obtained in Bianchini and Ravanini, Entanglement Entropy from Corner Transfer Matrix in Forrester Baxter non-unitary RSOS models, J. Phys. A: Math. Theor. 49 (2016) 154005.

And similarly in integrable non-unitary quantum spin chains here: Couvreur, Jacobsen and Saleur, Entanglement in non-unitary quantum critical spin chains, Phys. Rev. Lett. 119, 040601 (2017).

An alternative proposal was put forward in: Dupic, Estienne and Ikhlef, Entanglement entropies of minimal models from null-vectors, SciPost Phys. 4, 031 (2018).

Finally, formula (33) for the entanglement of a finite system, was also written in the earlier paper Holzhey, Larsen and Wilczek, Geometric and Renormalized Entropy in Conformal Field Theory, Nucl.Phys. B424 (1994) 443-467. (formula 2.13)

2) The interpretation of results is a little bit unclear to me, especially the fact that the duality the authors talk about seems to imply an exchange of space and momentum coordinates. Does this means that the types of entanglement that are obtained in the original and dual system are quite different? I mean, is the entanglement of the dual system still the entanglement associated to a bipartition in space? Or are we now talking about a bipartition in momentum space? 3) Depending on the answer to the question above, it might be relevant to explain what the entanglement entropy of the dual system actually means as a physical quantity. A standard point of view in non-hermitian systems, even those that can be mapped to hermitian ones, is that one can redefine the reduced density matrix by using the similarity transformation that relates the original non-hermitian hamiltonian to a hermitian counterpart. However, when doing so, what tends to happen is that the notion of space locality is not preserved. In other words if you for instance study a non-hermitian quantum spin chain and you find the counterpart of a Pauli matrix in the hermitian version of the model, you will generally find that the new operator now acts on many sites (so, it is not local anymore) and since the EE is an eminently local quantitiy (depends on a clear notion of bipartion) it is unclear what the EE of the hermitian version of the model actually means. An example of this can be seen for instance in the paper https://arxiv.org/abs/0906.4070. 4) More generally, it was not clear to me whether the duality the authors describe is the simple and well-known observation that a non-hermitian hamiltonian can be mapped to a hermitian (or non-hermitian) one via a similarity transformation. If it is just this, then I would say that this has been known and exploited in all the papers I pointed out above. If it goes beyond, then it would be worth emphasizing this more. 5) In the same spirit, a more direct questions is, are these operators R, P and their dual versions new operators in this context? More precisely, are the operators P, R unrelated to \Theta or even to Parity and Time reversal in PT-symmetric systems? I know that in the papers of Mostafazadeh that are cited it is eloquently argued that PT-symmetry is just a special case pseudo-hermiticity, but even so, because of the clear physical meaning of P and T transformations, PT symmetric systems are still widely studied. I think it will increase the impact of this work, if the authors could connect their results to such systems. For instance in a PT-symmetric system the parity operator P acts on the Hamiltonian just a Theta acts in equation (7).

  • validity: high
  • significance: high
  • originality: high
  • clarity: high
  • formatting: excellent
  • grammar: excellent

Author:  Peng Ye  on 2021-05-30  [id 1477]

(in reply to Report 1 on 2021-02-25)

Dear Referee,

We would like to thank you for reviewing our manuscript. Your constructive suggestions are of great help to revise our draft. Following the your kind report, we intend to modify the draft according to each.

Best regards

Li-Mei Chen, Shuai A. Chen and Peng Ye

---

## Round 1 · Referee Report · Anonymous · 2021-2-25

Strengths

1) The paper tries to systematize the treatment of non-unitary quantum mechanical systems in relation to computations of entanglement. This is a worthy effort, and I am not aware of other approaches were this is done so systematically. As such it opens a new pathway for research in this area.

2) The paper is quite well written and many examples are presented.

3) The paper elaborates further on the connection between non-unitarity and entanglement, establishing links between two areas of research that tend to be studied separately and from quite different viewpoints.

Weaknesses

1) The paper does not cite a lot of relevant literature, both in the context of non-hermitian quantum mechanics and that of entanglement in non-unitary systems.

2) The paper is not sufficiently clear about the novelty of some of the results, particularly the duality that the authors seem to highlight as one of their main contributions looks very similar to some standard results in the context of non-hermitian quantum mechanics.

3) Connected to the above, the authors should make a bigger effort to emphasize what is standard knowledge and what is new in their approach.

4) Finally, I have some questions regarding the interpretation of some of their results. I will elaborate further in my report.

Report

The referee writes:

In my view, the authors make an interesting contribution to two areas of study that have traditionally developed quite independently…. I summarize my main suggestions/questions in the "Requested changes" section.

Our Reply:

We thank for the Referee’s comments and recommendation.

The referee writes:

1) My first observation is that the paper only cites a very small number of works on non-hermitian quantum mechanics (references 33,34)…...

……..

Our Reply:

We sincerely thank the referee for the very helpful literature suggestions.

What we plan to revise:

The reference listed  will be updated by taking care of above suggested papers.

The referee writes:

2) The interpretation of results is …... a bipartition in momentum space.

Our Reply:

We thank the referee for letting us have opportunity to clarify the duality process. The exchange of space coordinates and momenta is just one piece of (the second step in Fig.1) the whole duality process. To make everything clear, let us go back to Fig. 1 and focus on the caption of Fig.1 and texts near Fig.1. Our duality establishes a connection between two distinct systems with the identical entanglement spectrum and entanglement entropy. Let us summarize the following several key points:

  1. In Fig.1, the duality is composed by two steps (① and ②).

  2. Preparation. Start with an original non-Hermitian free-fermion system $(H_\mathrm o)$. Its entanglement spectrum (ES) is uniquely determined by two projectors: $\mathcal P_\mathrm o$ and $\mathcal R_\mathrm o$ (See Eq. (11) and (12)). In other words, if we know the expressions of these two projectors, we can compute ES straightforwardly. More precisely, once real space partition (i.e., which spatial region will be traced out) is given, the expression of $\mathcal R_\mathrm o$ is fixed (See Eq. (13)). Once occupied single-particle states labeled by momenta are given, the expression of $\mathcal P_\mathrm o$ is also fixed (See Eq. (14)). For a Hermitian system, both two operators are Hermitian. For non-Hermitian systems, $\mathcal P_\mathrm o$ is non-Hermitian but $\mathcal R_\mathrm o$ must still be Hermitian by definition.

  3. Step-①. Then, we preform the first step (① in Fig.1). A similarity transformation is applied to both projectors. This transformation is unusual since it not only transforms $\mathcal P_\rm o$ defined momentum space but also transforms $\mathcal R_\rm o$ defined in the real coordinate space.
  4. Step-② In the second step, we do the “exchange”. But we need more careful definition on “exchange”. More precisely, we identify momenta in $\mathcal O^{-1}\mathcal P_\mathrm o\mathcal O$ as new coordinates and identify coordinates in $\mathcal O^{-1}\mathcal R_\mathrm o\mathcal O$ as new momenta. Immediately, we obtain two new projectors (denoted by $\mathcal P_\mathrm d$ and $\mathcal R_\mathrm d$ ) of a dual system whose Hamiltonian is $H_\rm d$, (see the arrows in Fig.1.)
  5. Since real space projector $\mathcal R_\mathrm d$ must always be Hermitian by definition, we need to constain $\mathcal O$ such that $\mathcal O^{-1}\mathcal P_\mathrm o\mathcal O$ is Hermitian and thereby $\mathcal R_\mathrm d$ is Hermitian.
  6. Based on the choice of $\mathcal O$, we can define two types of non-Hermitian models. In the newly added Table 1, we list key features of two types.

In conclusion, in both sides of duality, entanglement is always defined from reduced density matrix by partially tracing out real space.

What we plan to revise:

A new table (Table 1) will be added (see the attached file: file1 ).

The referee writes:

3) Depending on the answer to the question above, ……. An example of this can be seen for instance in the paper https://arxiv.org/abs/0906.4070.

4) More generally, …... then it would be worth emphasizing this more.

Our Reply:

We thank the referee for the insightful comment. We fully agree on the concern raised by the referee if the similarity transformation is defined in the way the referee mentioned, which is usually done in the so-called Dyson map.

However, our duality process is not defined in this way. In Fig.1, our similarity transformation, as also mentioned in above Reply, is simultaneously applied to act both quantum projectors, i.e., $\mathcal R_\rm o$ and $\mathcal P_\rm o$ in real space. This is the first step of the duality process, as shown in cartoon picture in Fig. 1.In a Dyson map, the similarity transformation is merely applied to $\mathcal P_\mathrm o$. Further, generally, a Dyson map alters entanglement spectrum while keeping energy spectrum invariant. In contrast, our duality preserves entanglement spectrum while it imposes no constraint on energy spectrum.

What we plan to revise:

At the end of Section-4, a paragraph will be added for explaining the above issue.

The referee writes:

5) In the same spirit, a more direct questions is, …… $\Theta $ acts in equation (7).

Our Reply:

We thank the referee for the suggestion. It is indeed a good idea to relate our duality theory to PT symmetric systems. The requirement in Eq. (7) excludes the possibility for choosing a parity operator as $\Theta$ for one cannot find a factorization $P=\mathcal{OO}^\dagger$. For this point, we will add a paragraph at the end of Sec 4.

The referee writes:

· Validity: High

· Significance: High

· Originality: High

· Clarity: High

· Formatting: Excellent

· Grammar: Excellent

Attachment:

file1.pdf

---

## Round 1 · Referee Report · Anonymous (Referee 2) · 2021-4-7

Strengths

  1. The paper reveals an interesting duality for the entanglement in non-Hermitian systems.
  2. Several examples try to make the duality clear.

Weaknesses

  1. Somewhat limited results for specific non-Hermitian systems.
  2. The discussion is concise and hard to follow at some places.

Report

The paper discusses an interesting duality for the entanglement of certain non-Hermitian systems. Though a bit limited in terms of applicability, I find it very appealing that the entanglement spectrum of a non-Hermitian system can be obtained from a corresponding Hermitian Hamiltonian, after exchanging real space and momentum space. In my opinion, the paper can be published in SciPost after some clarifications.

  1. First, it is not entirely clear what the distinction between type-I and II non-hermitian systems is. Some further explanation would be beneficial for the general reader.

  2. Non-hermitian Hamiltonian can possess real spectrum, and these are usually PT-symmetric in a general sense. I have the feeling that the transformation Theta stands for this. These non-Hermitian Hamiltonians can be mapped to Hermitian Hamiltonians by the transformation O from Eq. (8), if I understood the paper correctly. This Hermitian Hamiltonian has the same spectrum as the original Hamiltonian. Is it this hermitian Hamiltonian, whose entanglement properties are identical to the original non-Hermitian Hamiltonian?

In this context, I do not really understand Eq. (32), that t' should not be given for the entanglement properties. Why is it so? This also suggest that H_d is not the Hermitian counterpart of H_o after s similarity transformation since this would fix t'. Some more clarification on this would be really helpful.

  1. The authors discuss mainly real space entanglement, and their duality mapping involves exchanging real space and momentum space. However, one can also study momentum space entanglement ( arXiv:1603.01997 or arXiv:1404.7545) in the original model. Would the duality still work similarly in this case by exchanging real and momentum spaces?
  • validity: high
  • significance: high
  • originality: top
  • clarity: good
  • formatting: good
  • grammar: perfect

Author:  Peng Ye  on 2021-05-30  [id 1478]

(in reply to Report 2 on 2021-04-07)

Dear Referee,

We would like to thank you for reviewing our manuscript. Your constructive suggestions are of great help to revise our draft. Following the your kind report, we intend to modify the draft according to each.

Best regards

Li-Mei Chen, Shuai A. Chen and Peng Ye

Anonumous Report 2 on 2021-4-7 Invited Report

Strengths

  1. The paper reveals an interesting duality for the entanglement in non-Hermitian systems.
  2. Several examples try to make the duality clear.

Weakness

  1. Somewhat limited results for specific non-Hermitian systems.
  2. The discussion is concise and hard to follow at some places.

Report

The referee writes:

The paper discusses …... can be published in SciPost after some clarifications.

Our Reply:

We thank the Referee for the recommendation.

The referee writes:

  1. First, it is not entirely clear what the distinction between type-I and II non-hermitian systems is. Some further explanation would be beneficial for the general reader.

Our Reply:

We thank the referee for helping us how to revise the draft. In Section-3.2, we explain the physics of type-I and type-II. The idea of two-type classification is motivated by the observation that some non-Hermitian systems (dubbed “type-I”) can be dualized to Hermitian systems with the same entanglement. In this way, we conclude that the entanglement of type-I non-Hermitian systems (with a given real space partition) can be entirely reproduced by proper Hermitian systems by designing real space partition properly. If there is no way to obtain a Hermitian system after duality, we have type-II. As an example, we show, in Sec. 4.1, that the 1D nonreciprocal model (with half chain traced out) is of type-I. In Sec. 4.2, an example of type-II is discussed.

It is known that different designs of entanglement cut in real space may drastically alter the result of ES and EE. Thus, we emphasize that, to determine a non-Hermitian model is of type-I or type-II, we must be given how the real space is partitioned, which is encoded in the projector $\mathcal R_\mathrm 0$.

It is appealing to studying a finer structure of entanglement of a type-II system, which is beyond the scope of the present manuscript.

What we plane to revise:

We tend to add Table 1 (see attached file1) to summarize some basic properties of the two-type non-Hermitian systems. Eq. 24 is added for enhancing readability. More in-depth discussions on type-I and type-II are also planed to be added.

The referee writes:

  1. Non-hermitian Hamiltonian can possess real spectrum, and these are usually PT-symmetric in a general sense. I have the feeling that the transformation Theta stands for this. These non-Hermitian Hamiltonians can be mapped to Hermitian Hamiltonians by the transformation O from Eq. (8), if I understood the paper correctly. This Hermitian Hamiltonian has the same spectrum as the original Hamiltonian. Is it this hermitian Hamiltonian, whose entanglement properties are identical to the original non-Hermitian Hamiltonian? In this context, I do not really understand Eq. (32), that t' should not be given for the entanglement properties. Why is it so? This also suggest that H_d is not the Hermitian counterpart of H_o after s similarity transformation since this would fix $t^\prime$. Some more clarification on this would be really helpful.

Our Reply:

We thank the referee for letting us have opportunity to clarify the duality process and its physical consequences. To make everything clear, let us go back to Fig. 1 and focus on the caption of Fig.1 and texts near Fig.1. Our duality establishes a connection between two distinct systems with the identical entanglement spectrum and entanglement entropy. Let us summarize the following several key points:

  1. In Fig.1, the duality is composed by two steps (① and ②).

  2. Preparation. Start with an original non-Hermitian free-fermion system $(H_\mathrm o)$. Its entanglement spectrum (ES) is uniquely determined by two projectors: $\mathcal P_\mathrm o$ and $\mathcal R_\mathrm o$ (See Eq. (11) and (12)). In other words, if we know the expressions of these two projectors, we can compute ES straightforwardly. More precisely, once real space partition (i.e., which spatial region will be traced out) is given, the expression of $\mathcal R_\mathrm o$ is fixed (See Eq. (13)). Once occupied single-particle states labeled by momenta are given, the expression of $\mathcal P_\mathrm o$ is also fixed (See Eq. (14)). For a Hermitian system, both two operators are Hermitian. For non-Hermitian systems, $\mathcal P_\mathrm o$ is non-Hermitian but $\mathcal R_\mathrm o$ must still be Hermitian by definition.

  3. Step-①. Then, we preform the first step (① in Fig.1). A similarity transformation is applied to both projectors. This transformation is unusual since it not only transforms $\mathcal P_\rm o$ defined momentum space but also transforms $\mathcal R_\rm o$ defined in the real coordinate space.
  4. Step-② In the second step, we do the “exchange”. But we need more careful definition on “exchange”. More precisely, we identify momenta in $\mathcal O^{-1}\mathcal P_\mathrm o\mathcal O$ as new coordinates and identify coordinates in $\mathcal O^{-1}\mathcal R_\mathrm o\mathcal O$ as new momenta. Immediately, we obtain two new projectors (denoted by $\mathcal P_\mathrm d$ and $\mathcal R_\mathrm d$ ) of a dual system whose Hamiltonian is $H_\rm d$, (see the arrows in Fig.1.)
  5. Since real space projector $\mathcal R_\mathrm d$ must always be Hermitian by definition, we need to constain $\mathcal O$ such that $\mathcal O^{-1}\mathcal P_\mathrm o\mathcal O$ is Hermitian and thereby $\mathcal R_\mathrm d$ is Hermitian.
  6. Based on the choice of $\mathcal O$, we can define two types of non-Hermitian models. In the newly added Table 1, we list key features of two types.

After clarifying the duality process, let us go back to the Referee’s questions.

Our duality process keeps entanglement spectrum not energy spectrum invariant. Our duality is to building bridge between different models with different entanglement cuts, who share the same entanglement spectrum and entanglement entropy.

Therefore, our duality is not belonging to the usual category of duality techniques (e.g., Jordan-Wigner transformation) that keep energy spectrum invariant. A usual similarity transformation on Hamiltonians keeps physical energy spectrum invariant but cannot preserve entanglement spectrum.

More concretely, the duality in our work is not a naïve conventional similarity transformation on Hamiltonians, but a novel transformation (with two steps in Fig.1) on two projectors that uniquely determine entanglement spectrum and entanglement entropy. We recommend Section 2, especially Eqs. (11,12,13,14).

As a result, the dual $H_\mathrm d$  in Eq. (32) in the first version is constructed based on dual Fock-space projector $\mathcal P_\mathrm d$ that does not encode any information about physical energy spectrum.

What we plan to revise:

We tend to add a discussion on this point in the Conclusion section, “Significantly, our duality is typically distinct with a Dyson map. A Dyson is referred to a similarity transformation to relate to a non-Hermitian system to a Hermitian counterpart with an energy spectrum unchanged and it generally alters entanglement. Comparatively, our duality targets at entanglement while imposing no constraint on an energy spectrum.

The referee writes:

  1. The authors discuss mainly …... momentum spaces?

Our Reply:

We thank the referee for the suggestion. It is of great interest to generalize our duality to momentum-space entanglement. For a Hermitian system, such a duality is expected to work again for free fermion systems. For non-Hermitian system, we think the first step is to focus on definition of momentum space entanglement and then to check whether our duality works.

What we plan to revise:

We add a comment on this point in the Conclusion section. “Fourth, it is appealing to generalizing our duality for momentum space entanglement.”

The referee writes:

· Validity: High

· Significance: High

· Originality: Top

· Clarity: Good

· Formatting: Good

· Grammar: Perfect

Attachment:

file1_BpifOOx.pdf

---

## Round 2 · Referee Report · Anonymous · 2021-6-17

Strengths
Unchanged from first report
Weaknesses
No mayor weaknesses as all points I raised in my first report have been addressed.
Report
I am satisfied with the changes made by the authors as they have addressed all my queries in detail and improved the paper in several ways, including by expanding their set of references .
I am happy to accept the paper in its current form.

---

## Round 2 · Referee Report · Anonymous · 2021-6-21

Report
The authors have addressed my comments in detail. I find their response persuasive and think that their results are important and sound on the entanglement of certain non-hermitian systems. Therefore, I'd recommend publication in Scipost Physics.

---

## Round 2 · Author Response

Hereby we are resubmitting our work to SciPost Physics.
We thank you for sending us two referee reports. We have carefully read the reports and provided detailed replies, and especially point-to-point response. We thank two referees for their positive comments on scientific merit and creativity of our work. We made careful revision by following all constructive suggestions in the two reports.
Thanks a lot!
Bests

---

## Round 2 · List of Changes

We list main changes below.
1. Reference section was updated. The papers suggested by referees have been taken care of.
2. Abstract section was rewritten.
3. A useful table (Table 1) was added.
4. At the end of Section-4, a paragraph was added for explaining multiple issues.
5. Several typos were identified and corrected.
Description of changes is also available in our replies. Some important changes were marked in red in the revised draft.
Thanks!

---

## Editorial Decision

published